# Integrating Rock Dust and Organic Amendments to Enhance Soil Quality and Microbial Activity for Sustainable Crop Production

**DOI:** 10.3390/plants14081163

**Published:** 2025-04-09

**Authors:** Abraham Armah, Linda Alrayes, Thu Huong Pham, Muhammad Nadeem, Owen Bartlett, Eric Fordjour, Mumtaz Cheema, Lakshman Galagedara, Lord Abbey, Raymond Thomas

**Affiliations:** 1School of Science and the Environment, Grenfell Campus, Memorial University of Newfoundland and Labrador, Corner Brook, NL A2H 5G4, Canada; 2Biotron Experimental Climate Change Research Centre, Department of Biology, University of Western Ontario, London, ON N6A 3K7, Canada; 3Department of Plant, Food and Environmental Sciences, Faculty of Agriculture, Dalhousie University, Truro, NS B2N 5E3, Canada

**Keywords:** rock dust utilization, active microbial community, phospholipid fatty acid (PLFA) biomarkers, sustainable soil management

## Abstract

Rock dust (RD) is a by-product of the precious metal mining industry. Some mining operations produce close to 2,000,000 Mg of RD/year, posing disposal issues. This study evaluated the physicochemical and microbial properties of RD from gold mining and its potential use in RD-based growing media. Ten media formulations were tested: Promix (Control), 100% (RD), 100% topsoil (TS), 50% RD + 50% topsoil (RDT), 25% RD + 75% topsoil (RT), 50% RD + 50% Promix (RP), 50% RD + 25% biochar + 25% Promix (RBP), 50% RD + 25% compost + 25% Promix (RCP), 50% RD + 50% biochar (RB), and Huplaso (negative control). RD particle size ranged from 0.1 to 2 mm with a bulk density of 1.5 g cm^−3^, while RD-based media ranged from 0.8 to 1.1 g cm^−3^ showing increased porosity. Nutrient content was analyzed using Inductively Coupled Plasma Optical Emission Spectrometry (ICP-OES), and the active microbial community assessed using PLFA biomarkers via GC-MS/FID, *n* = 4 and *p* = 0.05. Microbial analysis identified five classes (protozoa, eukaryotes, Gram-positive (G+), Gram-negative (G−), and fungi (F)), with a significant increase in G−, G+, and F in RD-based amendment RBP (28%) compared to control P (9%). G+, G−, and F showed a strong negative correlation (r = −0.98) with pH, while calcium correlated positively (r = 0.85) with eukaryotes and a strong positive correlation (r = 0.95) of cation exchange capacity with G+. This study suggests blending RD with organic amendments improves physicochemical quality and microbial activity, supporting its use in crop production over disposal.

## 1. Introduction

The precious metal mining industry produces about 7–17 billion Mg of rock dust (RD) as by-products associated with detrimental effects if not handled properly [1]. For instance, RD from the precious metal mining industry is considered a waste by-product and is usually dumped on landfill sites. Such disposals of RD are significant environmental and financial concerns and can impact aquatic ecosystems, landscapes, and soil health [2] Therefore, finding sustainable management solutions for RD is crucial to mitigating its negative environmental impact and reducing disposal costs. Previous findings have revealed that RD contains several mineral nutrients in the required quantity to support plant growth, and there is a growing interest in assessing the potential of RD as a growing media substrate. However, existing studies present contradictory findings regarding its effectiveness as a nutrient source. Some studies report little to no effect of RD on crop growth [3], whereas others demonstrate that RD amendments successfully enhance plant growth and yield, such as in maize [4]. The beneficial effects of RD amendments have been attributed to its high concentration of magnesium (Mg), calcium (Ca), potassium (K), and phosphorus (P), as well as its ability to increase soil pH and cation exchange capacity (CEC) [5,6].

Food production is a big challenge in northern or boreal regions, including Newfoundland and Labrador province, due to agronomically poor soil quality (acidity, low fertility, stony and shallow soil) and extreme weather conditions [7]. Therefore, the development of alternative production systems to provide sustainable food supply is extremely needed. One approach could be taken by formulating the growing media to improve the soil quality using organic amendments in controlled environment conditions. RD as mining by-product can be used as organic amendments to achieve higher crop production because RD can improve the physicochemical and biological properties of growing media while minimizing the adverse effect on soil ecosystems. The precious metal mining in Newfoundland, Canada, produces over 2 M tons of RD per year as an industrial mining by-product. This amount can be a major source of cheap organic amendment while improving industrial waste management, which can be beneficial for sustainability of agriculture in Newfoundland [5].

The heavy metal content in RD, which can be harmful for the environment and human systems, is also of concern. However, ref. [8] reported analysis of heavy metal composition in RD from mining waste disposal that showed a very low concentration of arsenic, lead, mercury, nickel, and natural radioactive heavy elements. The concentration of heavy metals in RD (e.g., mercury = 0.02 mg/kg dry soil) was below the safe limit established by the Canadian Council of Ministers of the Environment (CCME) [9] for agricultural soils, as well as the safe limit for biosolids. The low heavy metal level in RD makes it safe for use as media amendment for re-mineralizing podzolic soils and safe as growing media for crops intended for human consumption.

While previous studies have explored the effects of rock dust (RD) on soil properties and plant growth, there remains a significant gap in understanding the specific physicochemical characteristics of RD that influence its effectiveness as a growing media amendment. Additionally, the impact of RD on microbial community composition and abundance within growing media is not well understood. This study aims to address these gaps by evaluating the physicochemical properties of RD and RD-based growing media, and assessing the microbial community composition and abundance in these media, to determine their influence on media quality and suitability for plant growth. We hypothesize that blending RD from gold mining with organic growing media enhances the active microbial community in the rhizosphere and improves physicochemical properties, thereby increasing the media’s suitability for sustainable crop production.

## 2. Materials and Methods

### 2.1. Experimental Site, Collection of Samples, and Preparation

The experiment was conducted in a walk-in growth chamber located at the Boreal Ecosystem Research Facility, Memorial University, NL, Canada. RD was obtained from deposits at Anaconda Mining Company, Baie Verte, NL (49°57′42″ N, 56°07′23″ W). A 2-year study was carried out between May 2019 and May 2021. First, RD was collected from different transects across the collection site and chemically analyzed to determine elements of environmental concern (e.g., mercury, lead, and cyanide). After the chemical analysis and the safe levels were ascertained from the analysis, RD was used to formulate 10 growing media amendments (Table 1).

### 2.2. Media Formulation with Rock Dust (RD) Amendment

Ten different media formulations were prepared to include varying ratios of RD amendment, as shown in Table 1. The Promix potting media, obtained from Premier Horticulture Inc. (Quakertown, PA, USA), consists of 75–85% sphagnum peat moss, vermiculite, wetting agent, dolomitic and calcitic limestone, and horticultural-grade perlite. Compost, Huplaso, Biochar, and topsoil were sourced from St-Isidore Asphalte Ltée (Saint-Isidore, NB, Canada). The RD mine tailing waste was provided by Anaconda Mining Inc., located in the Baie Verte Mining District (49°57′42′′ N, 56°07′23′′ W), north-central Newfoundland, Canada. Huplaso and biochar are alkaline, low-density amendments; Huplaso is mineral-rich, while biochar offers a high surface area and cation exchange capacity. Topsoil is a loamy, mineral-based material with neutral pH, moderate organic matter, and balanced physical properties conducive to plant growth. RD is a mineral-rich by-product of the mining industry, containing calcium and trace elements such as iron, aluminum, and magnesium, but low in nitrogen, phosphorus, and potassium. It has potential as a soil amendment to improve fertility, especially in nutrient-poor soils, and can support plant growth.

### 2.3. Media Preparation

The soil and other media were air-dried at room temperature for 48 h. A 1000 mL volumetric flask was used to measure and standardize the quantity of media components on a volume basis. A total volume of 1000 mL was defined as 100% composition, and ten media formulations were prepared according to their respective mix ratios. Each treatment medium was thoroughly homogenized manually to ensure uniform distribution. The prepared media mixtures were then transferred into 12-inch plastic planting pots for transplantation.

### 2.4. Particle Size Analysis

Particle size distribution analysis was performed using the wet sieve method as described by [10], with slight modifications. A sieve set conforming to ASTM standards was utilized, with sieve sizes ranging from 500 μm (largest) to 45 μm (smallest). The top coarse screen had a 425 μm opening, while each successive sieve had an aperture size half that of the one above, resulting in a 212.5 μm opening for the second sieve. Particles exceeding 2 mm in diameter were removed, and the remaining sample was subjected to wet sieving.

A total of 100 mL of distilled water was used to wash the soil, and the water passing through the sieve set was collected at the bottom. Soil particles retained on each sieve were carefully transferred into aluminum pans, oven-dried at 105 °C, and allowed to cool before weighing. The percentage weight of each particle size fraction was then calculated and used to construct a cumulative particle size distribution curve, based on the average of three replicates.

### 2.5. Physical Soil Properties Analysis

The basic physical properties of RD, including bulk density, porosity, and field capacity, were determined following the method described by [11].

To measure bulk density, three replicate empty metal rings were weighed (M1), then uniformly filled with the media by tapping three times on a solid surface to ensure compaction. The mass of the filled metal ring, including filter paper and a rubber band, was recorded (M2).

For field capacity determination, the samples were gradually saturated from the bottom over two days. The filled rings were placed inside an aluminum container, where water was initially added to cover one-third of the sample height. After approximately 8 h, water was increased to cover two-thirds, and finally, the water level was raised just below the brim of the metal ring. Saturation was considered achieved when a thin water film appeared on the sample surface. The saturated sample was then weighed (M3) before being transferred to a separate empty metal ring to drain freely under gravity for 1 to 3 days. Once drainage stabilized, the final weight (M4) was recorded. To prevent surface evaporation during drainage, the sample was covered with plastic wrap.

Samples were then oven-dried at 105 °C until a constant weight (M5) was reached. Bulk density (g/cm^3^) was calculated by dividing M5 by the volume of the sample. Porosity was determined using the formula (M3-M5)/sample volume, while field capacity was calculated as (M4-M5)/sample volume, assuming a water density of 1 g/cm^3^.

### 2.6. pH Analysis

Three replications of samples were taken from each of the formulated media amendments. In determining soil pH, the 1:2 method was adopted, as reported by [11]. The pH/EC/TDS/Temperature meter (HANNA-H19813-6 with CAL check, Oakville, ON, Canada) was used. A quantity of 20 g of an air-dried soil sample from RD and RD-based amendment was diluted in 40 mL of deionized water (1:2 ratio) in 50 mL polypropylene tubes (VWR, Mississauga, ON, Canada). Each of the samples was stirred at 100 rpm for 1 h. Samples were left to settle for 30 min, then the pH of the supernatant was measured.

### 2.7. Mineral Composition Analysis

Soil samples were air-dried in a drying oven at 40 °C until fully dried, with a maximum drying duration of one week. Once dried, the samples were sieved under a fume hood using a 2.00 mm sieve. The processed samples were then sent to the Department of Fisheries, Forestry, and Agriculture Laboratory in St. John’s for mineral composition analysis.

The mineral composition analysis began with pH measurement in a 1:1 soil-to-deionized water suspension. If the measured pH was below 6.4, the pH of the Adam–Evans buffer solution was also recorded after mixing with the soil. The limestone requirement was determined based on pH values to neutralize soil acidity.

To determine soil organic matter content, the sample was first placed into a pre-weighed crucible, dried overnight in a 105 °C drying oven, then cooled and reweighed. The sample was subsequently transferred to a muffle furnace at 430 °C for 6 h. After cooling, the final crucible weight was recorded.

For elemental analysis, soil sample extraction was performed using the Mehlich-3 extraction method with pre-prepared stock solutions. The extracted samples were analyzed using an auto-analyzer (Prodigy Inductively Coupled Plasma Optical Emission Spectrometer, ICP-OES).

### 2.8. Active Microbial Community Analysis

Fatty acids, analyzed as methyl esters, were quantified using gas chromatography–mass spectrometry with flame-ionization detection (GC-MS/FID) using the PLFA (phospholipid fatty acid) analysis platform to determine the active microbial community in each media formulation as described by [12]. The retention times and mass spectra of each of the fatty acid methyl esters (FAMEs) were identified by comparison with known standard mixtures or pure PLFAs, following the method in [13]. Calibration curves of each FAME were generated using standards obtained from Larodan Lipids (Malmö, Sweden), including both *cis* and *trans* forms of PLFA 18:2ω6. Identification was performed using GC-MS analysis. Specific PLFAs were used as biomarkers to assess microbial presence and abundance, as outlined by [14]. Bacterial-derived PLFAs were further categorized into Gram-positive (G+) and Gram-negative (G−) bacteria. The PLFA 10 Me18:0 was used as a biomarker for actinomycete bacteria, and the PLFAs 18:2ω6c and 18:1ω9c were used as fungal biomarkers [13].

### 2.9. Statistical Analyses

Data were statistically analyzed using one-way analysis of variance (ANOVA) and Pearson’s correlation analysis in XLSTAT Premium (Addinsoft, New York, NY, USA). As all subsamples conformed to a normal distribution, the application of these methods was appropriate. The differences between treatment means were separated by Fisher’s least significant difference (LSD; α = 0.05). Pearson’s correlation analysis was performed to test the relationships between variables. Multivariate analysis such as principal component analysis (PCA) was performed to determine the effect of RD and media amendments on physicochemical properties of formulated growing media, nutrient, and active microbial community, and ANOVA was used to test the relationship between variables.

## 3. Results and Discussion

### 3.1. Particle Size Distribution of Rock Dust

Particle size significantly influences the nutrient release properties of rock dust (RD), as smaller particles increase the surface area available for chemical reactions, thereby enhancing nutrient solubility [15]. In our study, we analyzed the particle size distribution of RD across three sites and found that the majority of particles were less than 0.5 mm in diameter (Figure 1), more than 70% at Site 3, over 80% at Site 2, and over 99% at Site 1. These findings suggest that the fine particle sizes of RD in these sites could enhance nutrient release rates, potentially improving soil fertility and plant growth. This observation aligns with previous research indicating that finer rock particles can enhance nutrient release rates, thereby potentially improving soil fertility and plant growth [16].

### 3.2. Physical Characteristics of Formulated RD-Based Amendments

In this study, we observed that incorporating rock dust (RD) with other amendments such as topsoil, Promix, and biochar resulted in decreased bulk density and increased porosity compared to using 100% RD alone (Table 2). Notably, all media formulations, including 100% RD, exhibited bulk densities below the root-restriction threshold of 1.6 g/cm^3^, which is favorable for root growth.

The reduction in bulk density is attributed to the addition of materials with lower densities. For instance, a mixture of 75% topsoil and 25% RD had a bulk density of 0.85 g/cm^3^, whereas 50% topsoil with 50% RD resulted in 1.10 g/cm^3^, and 100% RD measured at 1.50 g/cm^3^. Similar trends were observed with Promix and biochar additions. Specifically, combining 50% Promix or 50% biochar with RD yielded bulk densities of 0.87 g/cm^3^ and 0.96 g/cm^3^, respectively, compared to 1.17 g/cm^3^ for a 25% Promix + 25% biochar + 50% RD mixture, and 1.50 g/cm^3^ for 100% RD.

The improvement in bulk density when combining RD with biochar can be ascribed to the porous structure and large specific surface area of both materials. These properties can potentially alter pore-size distribution, bulk density, and surface area, thereby impacting media structure and porosity [17].

Moreover, the porosity of all media formulations remained within the acceptable water-filled pore space of 60%, indicating suitable moisture content for soil respiration and nitrogen cycling. Enhanced porosity improves infiltration rates and soil structure properties, which are beneficial for plant growth [18]

Our results are in alignment with [11], who reported a significant increase in volumetric water retention and porosity with organic amendments in soil. It could be ascribed to the organic matter content of amendments, which reduces soil bulk density to permit root growth and enhances micro-porosity in soil. Bulk density serves as an indicator of soil health, influencing nutrient composition, porosity, water retention, infiltration, root depth/restriction, and microbial activity. Field capacity and porosity represent favorable soil conditions and the soil’s ability to retain water and minerals for plant uptake. At field capacity, the balance of water and air content in soil is ideal for crop growth, providing a suitable physical environment and serving as a medium for heat balance through evaporation and cooling. Soil water content is essential for plant biochemical reactions, aiding nutrient transportation and oxygen circulation for root development.

According to [19], the bulk density less than or equal to 1.3 g/cm^3^ is considered good; between 1.3 g/cm^3^ and 1.55 g/cm^3^ is fair; and greater than 1.8 g/cm^3^ is deemed to resist root growth. Therefore, based on bulk density and porosity, our RD-based amendments are considered good for root growth. In summary, the incorporation of amendments such as topsoil, Promix, and biochar with RD effectively reduces bulk density and increases porosity, thereby creating a more favorable environment for root growth and overall plant development.

### 3.3. Nutrient Composition of RD and the Formulated Amendments

The soil analysis demonstrated that RD contains a substantial amount of macro- and micro-nutrients, including phosphorus (P), potassium (K), calcium (Ca), magnesium (Mg), sulphur (S), zinc (Zn), copper (Cu), iron (Fe), boron (B), manganese (Mn), and aluminum (Al) (Table 3). This result supports the potential of RD and RD-based amendments as suitable media amendments to provide plants with adequate nutrients for survival. Previous studies outline the function of the minerals found in RD. For example, Ca and Mg are essential for crop growth and to support enzyme function and nutrient transportation; Mg is also responsible for photosynthetic activity, and boron (B) plays a role in cell wall and membrane integrity [20,21,22].

The nutrient availability in the growing media of this study can be attributed to the synergistic influence of amendments (Promix, compost, and biochar) combined with the inherently nutrient-rich RD. For example, biochar contains high carbon content, while compost, topsoil, and Promix^TM^ contain high organic matter content. These properties influence nutrient availability through organic matter decay and cation exchange capacity. The findings from this study concur with previous studies conducted by [23]. Their study concentrated on the physical, mineralogical, and chemical characterization of a volcanic-rock mining by-product and the use of black oats and maize trials in the greenhouse to assess the by-product’s potential use as a soil remineralizer. The results revealed that the RD amendment appeared to contain a significant amount of trace minerals and plant nutrients suitable to support plant growth or crop production.

### 3.4. Active Microbial Community in RD and RD-Based Amendments

In this study, we employed PLFA analysis to determine quantitatively and qualitatively the active microbial community in all formulated growing media. Our findings reveal that RD and RD-based amendments contain a significant amount of diverse, active microbial biomass (Table 4). G+ and G− are found to be highest across all treatments compared to eukaryotes, protozoa, and fungi, except for Huplaso, which shows a higher fungi population. Interestingly, growing media RBP, RCP, RDT, RT, and RB showed notable increase in G+ plus G− bacterial communities in comparison with control growing media P. This could be attributed to different combinations of media with different physicochemical properties providing favorable conditions for the growth of the active microbial community [24]. Biochar, for example, may improve soil physical structure, including aeration and water retention; increase soil nutrient concentrations and pH; and provide macroporous environments that promote microbial colonization [25,26]. These observations are congruent with the experiment conducted by [27], who revealed that soil microbial composition was influenced by increased nutrient availability in Arctic tundra soils. Moreover, the G− plus G+ plus F in RD-based amendment RBP (28.36 ± 4.88%) significantly increase as compared to the control P (9.49 ± 0.06%). This could be ascribed to the clay minerals, organic carbon content, and organic matter supplied by the mixture of RD, biochar, and Promix^TM^, respectively.

PC1 and PC2 displayed 60.95% and 24.57% of the total variability, respectively (Figure 2). The observation plot depicted distinct segregation of the media amendment based on the centroid where RDT and RB amendments were grouped in quadrant Q-4 of the observation plot (Figure 2A). Media RBP and H are clustered in quadrant Q-1; TS, RD, P, RCP, RT RP, and P are clustered in quadrant Q-3. The biplot displays the relationship among the different variables where eukaryotes, G−, and G+ plus G− bacterial were grouped with RDT and RB growing media. The presence of microbial communities in RD and RD-based amendments could indicate good soil health, as stated by [28]. The findings from our study demonstrate that RD-based amendments promoted the growth and diversification of active microbial communities, possibly serving as a good indicator of RD-based media quality and potential as a suitable media amendment for crop growth.

### 3.5. Relationship Between RD Media Amendments, Physical Properties, Nutrient Composition, and Microbial Biomass

Our results from the PCA revealed the relationship between RD’s physical properties, RD-based amendment, and microbial community (Figure 3). The treatments RB, RD, RP, and RCP were clustered in quadrant 4 of the PCA; treatments RDT, RT, P, TS, and RT were clustered in quadrant 3; and treatments RBP and H separated into quadrant 1 and 2, respectively. The control (P) and topsoil (TS) clustered with other treatment in quadrant 3; however, the negative control (H) segregated into quadrant 2. Pearson correlation analysis was conducted to understand the above relationship better. The results (Table 5) exhibit a strong negative correlation (r = −0.982) of G+ plus G− plus F with soil pH, positive correlation (r = 0.62) of mineral Fe with G+ plus G− plus F, and a strong positive correlation of soil porosity with G+ (r = 0.72) and with Zn (r = 0.63). We also observed a strong positive correlation (r = 0.85) of Ca with eukaryotes and G− bacteria. At the same time, potassium (K) showed positive correlation with G+ bacteria (r = 0.58) and with protozoa (1P) (r = 0.74).

The results from this study improve our understanding of the relationship of RD-based amendments with soil microbial composition, physicochemical properties, and nutrient availability, which influence the overall quality of media to support crop growth. Previous studies have revealed that the soil’s physicochemical properties affect the microbial structure of the soil [29]. Soil properties have been reported to impact the microbial community, directly and indirectly, impacting plant abiotic and biotic stress via microbial diversity, composition, and network interactions [30]. Furthermore, ref. [30] demonstrated that soil aggregates of 2–4 mm and 1–2 mm had high respiration and porosity, respectively, influencing soil microbial community.

A closer look at the active microbial community in this study confirms that RD-based amendment has an impact on microbial composition. Conversely, the microbial composition of the media amendments influence mineral disaggregation and dissolution by excreting organic acids and carbon dioxide, which react with water to generate carbonic acid and dissolve calcium-rich rocks slowly to release nutrients available in RD [31]. The positive correlation of the media nutrient composition (Ca, Zn, Fe) with the microbial composition suggests that RD-based amendment could create an environment favorable for active soil microbes, which mineralized these elements from the RD-based media amendment. Overall, the RD-based media formulated appears to have suitable quality to support soil microbial growth that enhances nutrient mobilization and availability that may be adequate for plant growth, development, or crop production.

These findings align with previous studies highlighting the benefits of rock dust applications in agriculture. Research indicates that rock dust enhances plant–soil interactions and supports sustainable agricultural practices by reducing the need for frequent fertilizer applications through its slow nutrient release [4].

However, limitations and challenges remain in the use of RD as a soil amendment. The nutrient release rate from rock dust can be slow and may not meet the immediate nutrient demands of crops. Factors such as soil pH, microbial activity, and environmental conditions significantly influence the effectiveness of rock dust amendments. Furthermore, the sourcing, processing, and transportation of rock dust can be energy-intensive and economically burdensome, potentially offsetting some of its environmental benefits. These factors must be carefully considered when evaluating RD’s practical application in large-scale agricultural systems.

## 4. Conclusions

Our study demonstrated that the particle size of RD evaluated from precious metal mining waste is less than 2 mm, with the size distribution of 40% sand, 30% clay, and 30% silt. This texture is similar to clay loam soil, which is suitable for agriculture due to its capability to hold water and nutrients. Furthermore, RD-based amendments were found to have low bulk density and high porosity, which indicate its potential to retain moisture content for soil respiration and nitrogen cycling, enhancing their infiltration level and allowing root development of plants. RD has a similar particle size as clay and silt, demonstrating its ability to store and hold cations which support the enhancement of nutrient availability and fertility of RD-amended media. Furthermore, the analysis of the RD samples revealed the availability of several nutrients essential for crop growth and production. For example, a high amount of calcium (Ca) was observed in the media formulation RP > RB > RD = RCP, while we also observed a high amount of Mg in the media amendments, as follows: RP > RBP > RCP > RD. The active microbial community found in RD-based amendments appears to be responsible for the weathering of RD to release the nutrients available for plant growth. For instance, strong significant and positive correlations were observed between Fe, K, Ca, and G− plus G+ plus F.

In summary, the results depicted that RD amendments in different growing media resulted in improved growing media quality, including lower bulk density, higher porosity, improved soil microbial communities, and improved nutrient status compared to non-RD amendments. Overall, the study demonstrated the potential of RD as a suitable media amendment for crop production under controlled environmental conditions. This could be a strategy used to provide a sustainable solution for by-product disposal and serve as an alternative soil mineralizer in the horticultural or agricultural industries.

## Figures and Tables

**Figure 1 plants-14-01163-f001:**
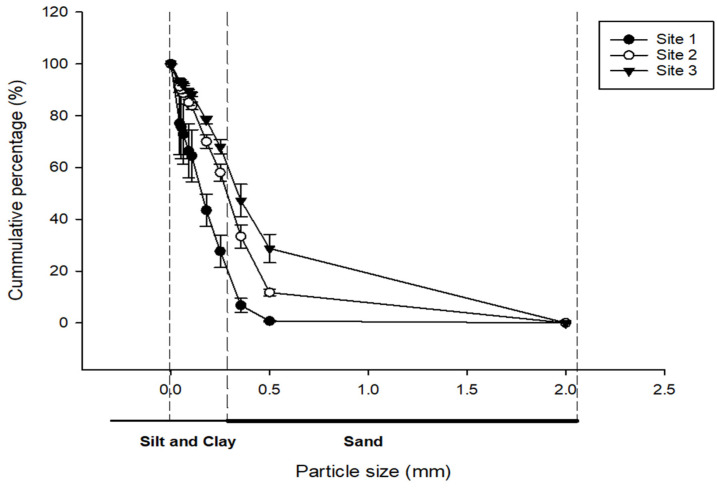
Particle size distribution of rock dust (RD) samples from three mining site. Y-axis value represents the average (*n* = 4) percentage of a particular size per site.

**Figure 2 plants-14-01163-f002:**
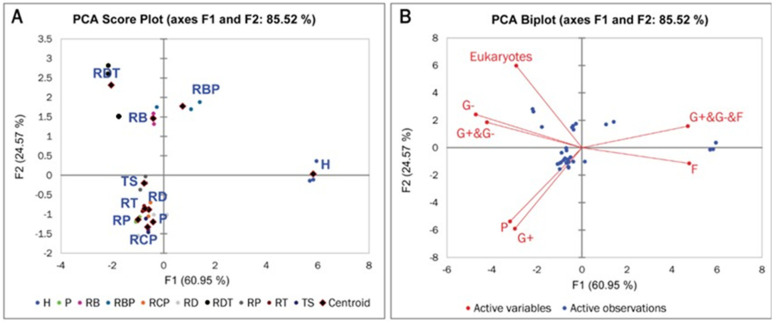
(**A**) Score plot and (**B**) biplot from principal component analysis (PCA) showing clusters of the RD-based media amendments based on the soil microbial community composition. Centroids of the 10 media formulations are indicated by diamonds, *n* = 4 per treatment. P = 100% Promix (positive control), RBP = 50% rock dust + 25% biochar + 25% Promix. H = Huplaso (negative control), RT = 25% rock dust + 75% topsoil, RP = 50% rock dust + 50% Promix, RCP = 50% rock dust + 25% compost + 25% Promix, RD =100% rock dust (by-product), RDT = 50% rock dust + 50% topsoil, RB = 50% rock dust + 50% biochar, TS = 100% topsoil.

**Figure 3 plants-14-01163-f003:**
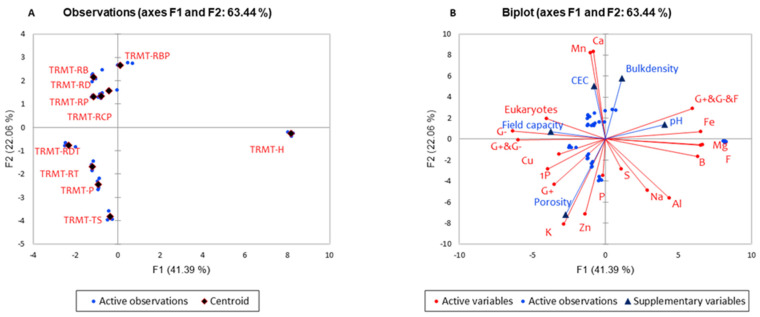
Principal component analysis (PCA) of physical and chemical composition of rock dust-based amendment: (**A**) Observation; (**B**) Biplot. P = 100% Promix (positive control), RBP = 50% rock dust + 25% biochar + 25% Promix. H = Huplaso (negative control), RT = 25% rock dust + 75% topsoil, RP = 50% rock dust + 50% Promix, RCP = 50% rock dust + 25% compost + 25% Promix, RD = 100% rock dust (by-product), RDT = 50% rock dust + 50% topsoil, RB = 50% rock dust + 50% biochar, TS = 100% topsoil.

**Table 1 plants-14-01163-t001:** Treatment of RD and RD-based media formulations with code and the combination level.

Treatment Code	Treatment	Combination Level
P	Promix^TM^	100%
RBP	RD + biochar + Promix^TM^	50% + 25% + 25%
H	Huplaso	100%
RT	RD + topsoil	25% +75%
RP	RD + Promix	50% + 50%
RCP	RD + compost + Promix^TM^	50% + 25% + 25%
RD	RD	100%
RDT	RD + topsoil	50% + 50%
RB	RD + biochar	50% + 50%
TS	Topsoil	100%

**Table 2 plants-14-01163-t002:** The physical composition of rock dust and rock dust-based amendments.

Treatment	Bulk Density (g/cm^3^)	Porosity (%)	Field Capacity (%)
P	0.24 ± 0.06 ^a^	60.69 ± 0.02 ^c^	30.52 ± 0.04 ^bc^
RBP	1.17 ± 0.01 ^g^	44.00 ± 0.01 ^a^	35.90 ± 0.01 ^c^
H	1.19 ± 0.01 ^f^	41.07 ± 0.01 ^a^	22.62 ± 0.01 ^a^
RT	0.85 ± 0.01 ^c^	56.44 ± 0.01 ^b^	35.15 ± 0.01 ^c^
RP	0.87 ± 0.01 ^c^	55.87 ± 0.01 ^b^	36.29 ± 0.01 ^c^
RCP	0.93 ± 0.01 ^e^	59.85 ± 0.02 ^bc^	36.09 ± 0.03 ^c^
RD	1.50 ± 0.01 ^h^	43.92 ± 0.02 ^a^	26.74 ± 0.03 ^c^
RDT	1.10 ± 0.01 ^f^	43.21 ± 0.01 ^a^	30.40 ± 0.01 ^bc^
RB	0.96 ± 0.01 ^d^	45.74 ± 0.01 ^a^	31.84 ± 0.01 ^a^
TS	0.57 ± 0.00 ^b^	71.66 ± 0.04 ^d^	32.11 ± 0.01 ^bc^
Recommended level	<1.6	50–70	10–33

Values represent mean ± standard error. Means in the same row accompanied by different superscripts are significantly different at LSD; α = 0.05, *n* = 4 per experimental replicate. P = 100% Promix (positive control), RBP = 50% rock dust + 25% biochar + 25% Promix. H = Huplaso (negative control), RT = 25% rock dust + 75% topsoil, RP = 50% rock dust + 50% Promix, RCP = 50% rock dust + 25% compost + 25% Promix, RD = 100% rock dust (by-product), RDT = 50% rock dust + 50% topsoil, RB = 50% rock dust + 50% biochar, TS = 100% topsoil.

**Table 3 plants-14-01163-t003:** Quantity of extractable minerals and pH of rock dust and rock dust-based amendments.

Minerals (mg kg^−1^)	RD	TS	H	P	RP	RB	RT	RDT	RCP	RBP	Level Required for Crop Growth
Phosphorus	1 ^a^	109 ^f^	4 ^ab^	89 ^e^	15 ^cd^	31 ^d^	86 ^e^	76 ^e^	25 ^cd^	15 ^bc^	12–14
Potassium	35 ^a^	201 ^c^	101 ^b^	387 ^e^	254 ^d^	13 ^a^	44 ^a^	29 ^a^	152 ^b^	116 ^b^	121–181
Calcium	9747 ^f^	3905 ^j^	4909 ^e^	2435 ^a^	11,102 ^j^	10,497 ^ij^	4102 ^d^	3029 ^bc^	9470 ^f^	10,276 ^h^	>40–60
Magnesium	126 ^b^	175 ^e^	517 ^h^	217 ^g^	205 ^f^	132 ^c^	130 ^b^	104 ^a^	217 ^i^	169 ^d^	50–70
Sulphur	40 ^g^	40 ^g^	34 ^de^	37 ^f^	33 ^de^	22 ^a^	30 ^c^	26 ^b^	31 ^cd^	37 ^ef^	10–20
Zinc	0.4 ^a^	20.2 ^i^	1.6 ^d^	1.4 ^cd^	2.9 ^f^	2.1 ^ef^	12.6 ^h^	7.8 ^g^	1.3 ^c^	0.9 ^b^	>1.5
Copper	1.6 ^c^	2.7 ^d^	0.9 a^b^	0.6 ^a^	2.7 ^de^	2.7 ^de^	2.8 ^e^	3.2 ^f^	0.9 ^ab^	1.1 ^b^	1–1.8
Sodium	11 ^a^	110 ^f^	114 ^g^	86 ^e^	45 ^c^	78 ^d^	11 ^a^	75 ^d^	21 ^b^	21 ^b^	-
Iron	396 ^c^	236 ^b^	1175 ^d^	73 ^a^	259 ^b^	241 ^b^	248 ^b^	258 ^b^	249 ^b^	255 ^b^	6.0–1 × 10^6^
Boron	1.2 ^c^	1.7 ^d^	4.3 ^e^	0.3 ^a^	0.5 ^ab^	0.4 ^ab^	0.9 ^bc^	0.6 ^ab^	0.3 ^a^	0.3 ^bc^	1–3
Manganese	147 ^f^	73 ^c^	77 ^d^	6 ^a^	174 ^j^	163 ^h^	72 ^b^	83 ^e^	148 ^g^	166 ^h^	20–30
Aluminum	38 ^b^	439 ^d^	626 ^e^	41 ^b^	7 ^a^	24 ^ab^	435 ^d^	228 ^c^	12 ^a^	17 ^ab^	<2–5
pH	8.4 ^de^	7.2 ^f^	9.1 ^g^	4.9 ^a^	7.4 ^c^	8.0 ^de^	8.0 ^ef^	7.1 ^ef^	6.5 ^b^	7.4 ^cd^	5.5–7.0

Mineral composition and pH level of rock dust, rock dust-based amendment, topsoil, compost and Promix^TM^. The last column shows the required level of nutrient and pH for crop growth. P = 100% Promix (positive control), RBP = 50% rock dust + 25% biochar + 25% Promix. H = Huplaso (negative control), RT = 25% rock dust + 75% topsoil, RP = 50% rock dust + 50% Promix, RCP = 50% rock dust + 25% compost + 25% Promix, RD = 100% rock dust (by-product), RDT = 50% rock dust + 50% topsoil, RB = 50% rock dust + 50% biochar, TS = 100% topsoil. Means in the same row accompanied by different letters are significantly different at LSD; α = 0.05, *n* = 4 per experimental replicate.

**Table 4 plants-14-01163-t004:** The microbial composition of rock dust and rock dust-based media amendments.

Treatment	Eukaryotes (%)	G− (%)	G+ (%)	F (%)	P (%)	G+ plus G− (%)	G+ plus G− plus F (%)
P	3.52 ± 0.09 ^d^	21.72 ± 0.39 ^def^	26.40 ± 0.25 ^def^	9.33 ± 0.13 ^c^	3.30 ± 0.08 ^e^	26.25 ± 0.02 ^b^	9.49 ± 0.06 ^c^
RBP	4.94 ± 0.35 ^f^	18.45 ± 0.93 ^b^	9.71 ± 0.73 ^ab^	7.12 ± 0.57 ^b^	1.44 ± 0.11 ^ab^	29.97 ± 2.21 ^c^	28.36 ± 4.88 ^d^
H	0.00 ± 0.00 ^a^	2.99 ± 0.25 ^a^	7.51 ± 0.05 ^a^	40.16 ± 0.13 ^g^	0.77 ± 0.03 ^a^	18.33 ± 0.16 ^a^	30.49 ± 0.16 ^e^
RT	1.37 ± 0.03 ^b^	19.91 ± 0.38 ^bc^	22.48 ± 0.11 ^d^	11.76 ± 0.17 ^f^	2.55 ± 0.07 ^ab^	36.06 ± 0.26 ^e^	5.89 ± 0.17 ^ab^
RP	4.21 ± 0.10 ^e^	23.65 ± 1.15 ^d^	27.24 ± 0.36 ^ef^	10.01 ± 0.23 ^cde^	1.98 ± 0.09 ^bc^	25.43 ± 0.07 ^b^	7.49 ± 0.17 ^abc^
RCP	2.15 ± 0.08 ^c^	19.89 ± 0.23 ^bc^	23.28 ± 0.41 ^de^	12.67 ± 0.06 ^f^	2.68 ± 0.10 ^d^	31.81 ± 0.41 ^cd^	7.52 ± 0.07 ^ab^
RD	1.98 ± 0.11 ^bc^	22.00 ± 1.09 ^cd^	26.39 ± 0.39 ^def^	9.45 ± 0.32 ^cd^	2.79 ± 0.25 ^de^	24.49 ± 0.52 ^b^	12.91 ± 2.49 ^c^
RDT	8.27 ± 0.45 ^g^	31.16 ± 1.33 ^e^	17.00 ± 3.97 ^c^	5.21 ± 0.39 ^a^	1.42 ± 0.19 ^ab^	34.15 ± 1.55 ^de^	2.78 ± 0.20 ^a^
RB	3.97 ± 0.11 ^de^	23.70 ± 1.10 ^d^	12.01 ± 0.31 ^b^	10.38 ± 0.13 ^de^	1.41 ± 0.08 ^ab^	36.91 ± 0.46 ^e^	11.61 ± 0.12 ^bc^
TS	1.67 ± 0.04 ^bc^	17.78 ± 1.13 ^b^	30.19 ± 0.46 ^f^	10.44 ± 0.20 ^e^	2.30 ± 0.05 ^cd^	31.38 ± 0.49 ^cd^	6.25 ± 0.02 ^ab^

Different letters show significant differences between treatments. Means in the same row accompanied by different superscript are significantly different at LSD; α = 0.05, *n* = 4 per experimental replicate. P = 100% Promix (positive control), RBP = 50% rock dust + 25% biochar + 25% Promix. H = Huplaso (negative control), RT = 25% rock dust + 75% topsoil, RP = 50% rock dust + 50% Promix, RCP = 50% rock dust +25% compost + 25% Promix, RD = 100% rock dust (by-product), RDT = 50% rock dust + 50% topsoil, RB = 50% rock dust + 50% biochar, TS = 100% topsoil.

**Table 5 plants-14-01163-t005:** Correlation between physicochemical and microbial properties of rock dust (RD) amended media.

**QUADRANT 1** **(RBP)**	Variables	pH	G+ plus G− plus F				
	pH		−0.982				
	G+ plus G− plus F	−0.982					
	Fe		0.618				
**QUADRANT 2 (H)**	Variables	F	Na				
	Na	−0.997					
	Al		0.958				
**QUADRANT 3 (RDT, RT, P, TS)**	Variables	Porosity	G+	1P	G+ plus G−	K	Zn
	G+	0.715					
	G+ plus G−		−0.574	−0.575			
	P	0.680					
	K		0.580	0.737	−0.885		
	Zn	0.634					
	Cu			−0.822	0.876	−0.902	0.558
**QUADRANT 4 (RB, RD, RP, RCP)**	Variables	CEC	Eukaryotes	G−			
	G−		0.585				
	G+	0.949					
	Ca		0.847	0.579			

The media compositions are in italics: Bulk density, porosity, field capacity, eukaryotes, G− = gram negative bacteria, G+ = gram positive bacteria, F = fungi, 1P = protozoa, G+ plus G− = gram positive and gram negative bacterial, pH = soil pH, CEC = cation exchange capacity, P = phosphorus, K = potassium, Ca = calcium, Mg = magnesium, S = sulphur, Zn = zinc, Cu = copper, Na = sodium, Fe = iron, B = boron, Mn = manganese, Al = aluminum. P = 100% Promix (positive control), RBP = 50% rock dust + 25% biochar + 25% Promix, H = Huplaso (negative ICP), RT = 25% rock dust + 75% topsoil, RP = 50% rock dust + 50% Promix, RCP = 50% rock dust + 25% compost + 25% Promix, RD = 100% rock dust (by-product), RDT = 50% rock dust + 50% topsoil, RB = 50% rock dust + 50% biochar, TS = 100% topsoil. *n* = 4. Values in bold are different from 0 with a significance level alpha = 0.05.

## Data Availability

The original contributions presented in this study are included in the article. Further inquiries can be directed to the corresponding authors.

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
