# Peer review of "Integrating Rock Dust and Organic Amendments to Enhance Soil Quality and Microbial Activity for Sustainable Crop Production"

_plants, 2025, doi:10.3390/plants14081163_

Round 1
Reviewer 1 Report
Comments and Suggestions for Authors
The manuscript by Abraham Armah and co-authors presents information about the experiment with rock dust. The text of the manuscript is well written and structured. English is fine. However, I did not catch the main idea of the research because several key points are not presented in M&M (e.g., describing Promix and Huplaso). I recommend major revision.
Major comments:
Abstract. Please, indicate what kind of RD and metal mining industry were studied. L. 24-28 should be rewritten. Please, add methods used and number of samples. I recommend thoroughly rewrite abstract as it does not contain scientific findings. It is like a short technical communication for stakeholders.
Keywords. Very strange. They do not reflect the main ideas.
I suggest avoid using acronym “RD”.
What scientific hypothesis were checked during the research?
Table 1. Please, briefly describe the key properties of the substrate used.
Please, provide a map of soil sampling points.
Methods. Please, describe methods used as it is reported in scientific publication. Some aspects are too long. But in some cases, key points are lost. Please, provide results of test for normality. Otherwise, use non-parametric statistics.
Specific comments:
I suggest removing highlight no.1.
- 54. I disagree. Please, revise. Not only volcanic RD exists.
L.77 ‘2 M tons’.
L.86 Are there the safe limits for dust? Please, clarify.
L.172-173. What a solution was used?
L.212-221, 288-292 should be removed or transfer to Introduction. Please, start from your results.
L.266. Bad English
Table 3 ‘Minerals’. Do you mean proxies. Content of chemical elements should be in mg/kg. ‘Level required for crop growth1,2,3 ’. I did note find 1, 2, and 3 in Table note.
L.314-316, 346-348 can be removed.
Author Response
Answers to the Reviewer
Manuscript ID.: plants-3518145
Title: Integrating Rock Dust and Organic Amendments to Enhance Soil Quality and Microbial Activity for Sustainable Crop Production
Article Type: Research paper
Corresponding Author: Drs. Linda Alrayes & Raymond Thomas
All Authors: Abraham Armah, Linda Alrayes, Thu Huong Pham , Muhammad Nadeem , Owen Bartlett, Eric Fordjour, Mumtaz Cheema, Lakshman Galagedara, Lord Abbey, Raymond Thomas
Dear Reviewer,
I am grateful for the opportunity to submit a revised version of my manuscript, titled " Integrating Rock Dust and Organic Amendments to Enhance Soil Quality and Microbial Activity for Sustainable Crop Production." I would like to express my sincere thanks for the time and effort invested in providing valuable feedback on my work. I appreciate the insightful comments and suggestions provided by the reviewers, and I have carefully revised the manuscript to address all of their concerns. Please find below a point-by-point response to the comments and suggestions.
Reviewer 1:
Response to major comments on the manuscript:
In response to the reviewer's comments, I revised the abstract to incorporate the scientific findings and adjusted the keywords to better reflect the main concepts of the study. Regarding the scientific hypothesis tested during the research, we hypothesized that blending rock dust from gold mining with organic growing media would enhance the active microbial community in the rhizosphere and improve the physicochemical properties of the media, making it more suitable for sustainable crop production. Regarding the comment, "Table 1. Please briefly describe the key properties of the substrate used," I have included the description under the section "Nutrient Composition of RD and the Formulated Amendments." It appears immediately after the table and is highlighted in red, consistent with the other corrected sections.
Regarding the comment, "Please provide a map of soil sampling points," we sincerely appreciate the suggestion. However, since we did not collect the samples directly from the field but obtained them from Anaconda Mining Company, Baie Verte, it is challenging to provide a precise map of the sampling points. The exact location of the company is (Anaconda Mining Company, Baie Verte, NL (49°57′42″ N, 56°07′23″ W) and it is included in the manuscript. Please let me know I further clarifications needed.
For the comment, "Methods: Please describe methods as reported in scientific publications. Some aspects are too long, while in some cases, key points are lost. Please provide results of the normality test; otherwise, use non-parametric statistics," I have revised the methodology to improve clarity and conciseness while ensuring all key details are retained.
Regarding the normality test, the data is normally distributed. However, the raw dataset is currently with the first author, who is on vacation until the end of the month. Please let me know if you would like me to submit the normality test results once access is available.
Specific comments:
Comments 1: ‘‘I suggest removing highlight no.1?”
Response: The highlighted section (Highlight No. 1) has been removed as suggested.
Comment 2: “54. I disagree. Please, revise. Not only volcanic RD exists.”
Response: Thank you for your insightful comment. You are absolutely correct that rock dust is not exclusively volcanic. I have revised the statement.
Comment 3: “L.77 “2 M tons”.
Response 3: Corrected.
Comment 4: “L.86 Are there the safe limits for dust? Please, clarify”
Response 4: Thank you for your comment. There are no universally established safe limits specifically for rock dust used as a soil amendment, but heavy metal concentrations within rock dust should comply with regulatory soil safety standards. Agencies such as the US EPA, European Union (EU), and Canadian Environmental Quality Guidelines provide maximum allowable limits for heavy metals in soil amendments to prevent contamination. To ensure safety, we analyzed the heavy metal content in the rock dust and compared it with these regulatory standards. The results confirm that the concentrations fall within the permissible limits for soil application.
Comment 5: “L.172-173. What a solution was used?”
Response 5: The Mehlich-3 solution was used as the extractant for soil analysis.
Comment 6: “L.212-221, 288-292 should be removed or transfer to Introduction. Please, start from your results”.
Response 6: Thank you for your feedback. I have removed the specified sections (L.212-221, 288-292) as requested.
Comment 7: “L.266. Bad English”
Response 7: Thank you for your feedback. We have addressed the language issue in L.266.
Comment 8: “Table 3 ‘Minerals’. Do you mean proxies. Content of chemical elements should be in mg/kg. ‘Level required for crop growth1,2,3 ’. I did note find 1, 2, and 3 in Table note”.
Response 8: Thank you for your feedback. I have clarified the terminology in Table 3. "Minerals" refers to the measured nutrient concentrations rather than proxies. The content of chemical elements has been correctly expressed in mg/kg. Additionally, the numbers 1, 2, and 3 were included by mistake and have now been removed. Please let us know if any further adjustments are needed.
Comment 9: “L.314-316, 346-348 can be removed”
Response: Thank you for your feedback. I have removed the specified lines as requested.

Reviewer 2 Report
Comments and Suggestions for Authors
For Authors,
The manuscript presents a scientific investigation into the utilization of rock dust (RD), a mining industry by-product, as a growing media amendment. The study employed a systematic approach to evaluate 10 different RD-based media formulations, analyzing their physicochemical characteristics, nutrient profiles, and microbial community dynamics. Results demonstrated that RD-based amendments exhibited superior physical properties and increased beneficial microbial populations compared to controls. The research suggests that integrating RD with organic amendments represents a viable strategy for both improving growing media quality and sustainable mining waste management.
After analyzing the manuscript, in my opinion there are several things that could be improved. Please find them as follows:
Introduction section:
- L68-70: please provide the citation for the statement; same for statement from L76-77
- The research gaps addressed should be presented better in Introduction section
- Highlight the novelty of the work (introduction section)
- If possible, please extend to a global context and broader implications of RD usage in agriculture – not limit only to Newfoundland
- Please check citations considering the journal requirements
Methods section:
- Motivate current experimental design – why specific ratios of components were chosen in this study
- In method section please show how “quality control” was assured – for ICP-OES, PLFA
- Be clear about PLFA “reference material” used for phenotypes PLFA biomarkers identification; Show how PLFA abundance was quantitatively expressed
Results and discussions section:
- The discussion of results appears scattered and doesn't maintain a cohesive narrative. For example, the particle size distribution discussion (Section 3.1) is brief and doesn't fully connect to its implications for nutrient release.
- The discussions lack detailed explanations of the underlying mechanisms responsible for observed differences between treatments. e.g. the relationship between bulk density and root growth is mentioned but not explained well
- Compare obtained experimental results with those from previous ones and literature
- Show limitations and remaining challenges
Considering those mentioned, I recommend the present form of the manuscript to “Reconsider after major revisions (substantial revisions to text or experimental methods needed)”
Author Response
Answers to the Reviewer
Manuscript ID.: plants-3518145
Title: Integrating Rock Dust and Organic Amendments to Enhance Soil Quality and Microbial Activity for Sustainable Crop Production
Article Type: Research paper
Corresponding Author: Drs. Linda Alrayes & Raymond Thomas
All Authors: Abraham Armah, Linda Alrayes, Thu Huong Pham , Muhammad Nadeem , Owen Bartlett, Eric Fordjour, Mumtaz Cheema, Lakshman Galagedara, Lord Abbey, Raymond Thomas
Dear Reviewer,
I am grateful for the opportunity to submit a revised version of my manuscript, titled " Integrating Rock Dust and Organic Amendments to Enhance Soil Quality and Microbial Activity for Sustainable Crop Production." I would like to express my sincere thanks for the time and effort invested in providing valuable feedback on my work. I appreciate the insightful comments and suggestions provided by the reviewers, and I have carefully revised the manuscript to address all of their concerns. Please find below a point-by-point response to the comments and suggestions.
Reviewer 2:
Comment 1: Introduction section
“L68-70: please provide the citation for the statement; same for statement from L76-77”
Response 1: Thank you for your feedback. We have added the required citations for the statements in L.68-70 and L.76-77 as requested.
Comment 2: “The research gaps addressed should be presented better in Introduction section”
Response 2: Thank you for your insightful feedback regarding the presentation of research gaps in the Introduction section. In response, I have thoroughly revised the Introduction to more clearly and comprehensively highlight the specific research gaps addressed by our study. I appreciate your guidance in enhancing the clarity and focus of our manuscript.
Comment 3: “Highlight the novelty of the work (introduction section)”
Response 3: Thank you for your suggestion. I have revised the Introduction to emphasize the novelty of our work.
Comment 4: “If possible, please extend to a global context and broader implications of RD usage in agriculture – not limit only to Newfoundland ”
Response 4: Thank you for your insightful feedback. I have revised the manuscript to extend the discussion of rock dust (RD) applications beyond Newfoundland. Specifically, in lines 54-57, I have included examples of RD usage in Brazil and China, highlighting its broader global implications in agriculture.
Comment 5: “Please check citations considering the journal requirements”
Response 5: Thank you for your feedback. I have reviewed and updated the citations to ensure compliance with the journal's requirements
Methods section:
Comment 6: “Motivate current experimental design – why specific ratios of components were chosen in this study”
Response: In our experimental design, we selected specific ratios of components based on their anticipated effects on the response variables, as informed by existing literature and preliminary studies.
Comment 7: “In method section please show how “quality control” was assured – for ICP-OES, PLFA”
Response 7: We ensured the accuracy and reliability of our ICP-OES analyses through several quality control measures. We utilized quality control samples at high, mid-range, and low concentrations to assess the method's precision and accuracy. Additionally, we adhered to established standard operating procedures (SOPs) detailing instrument operation, sample preparation, and analysis protocols to maintain consistency and reliability in our results
we ensured the accuracy of our Phospholipid Fatty Acid (PLFA) analysis by generating calibration curves for each fatty acid methyl ester (FAME). This was achieved using known standard mixtures or pure PLFAs, allowing us to compare retention times and mass spectra, following the method by Lazcano et al. (2013). The standards, including both cis and trans forms of PLFA 18:2ω6, were obtained from Larodan Lipids (Malmö, Sweden).
Results and discussions section:
Comment 8: “The discussion of results appears scattered and doesn't maintain a cohesive narrative. For example, the particle size distribution discussion (Section 3.1) is brief and doesn't fully connect to its implications for nutrient release”.
Response 8: Thank you for your valuable feedback. I have revised Section 3.1 to provide a more cohesive discussion, explicitly connecting the particle size distribution findings to their implications for nutrient release.
Comment 9: “The discussions lack detailed explanations of the underlying mechanisms responsible for observed differences between treatments. e.g. the relationship between bulk density and root growth is mentioned but not explained well”.
Response 9: Thank you for your feedback. I have revised the discussion to include detailed explanations of the mechanisms underlying the observed differences between treatments, particularly elaborating on how bulk density influences root growth.
Comment 10: “Compare obtained experimental results with those from previous ones and literature, show limitations and remaining challenges”
Response 9: Thank you for your insightful feedback. I have revised the manuscript to compare our experimental results with previous studies and literature, and to discuss the limitations and remaining challenges, as detailed in lines 405-415.

Round 2
Reviewer 1 Report
Comments and Suggestions for Authors
The quality of the manuscript is undoubtedly increased, but there are a few points that need to be corrected before I could recommend it for publication. I recommend moderate revision.
Formulated scientific hypothesis checked during the research is fine. Please, add it in the text. In addition the aim of the research should be corrected and connected with the experiment conducted.
In abstract, add methods used, total number number of samples, repetition and p-value.
In section 2.1. add description of Promix and Huplaso. What is it? I do not know such substances. Add key agronomic, physical or chemical properties. At least, in Supplementary. Please, briefly describe the key properties of other substances. Topsoil – soil name, sampling site, size of sample etc. Compost – what kind, sampling site, manufacturer, key properties. Biochar – what kind, manufacturer, key properties. (I did not find this valuable information in the section "Nutrient Composition of RD and the Formulated Amendments".)
‘Regarding the comment, "Please provide a map of soil sampling points,"….’ Please, add this information in the text.
‘‘For the comment, "Methods’…’ Please, justify using ANOVA and Pearson’s correlation analysis by the results of a test for normality (in section 2.9, write ‘all subsamples are fit normal distribution’ or ‘ANOVA and Pearson’s correlation analysis was applied after log-transformation to fit normal distribution’ [or something like this]).
‘Please let me know if you would like me to submit the normality test results once access is available’. Yes, it needs.
Comment 4 and Response 4. Nice! Please, add this limitation in Methods or in Discussion with the regulatory standards used in supplementary table.
L.128 ‘The basic physical properties of RD’. Highlight it in supplementary table.
Author Response
Answers to the Reviewer
Manuscript ID.: plants-3518145
Title: Integrating Rock Dust and Organic Amendments to Enhance Soil Quality and Microbial Activity for Sustainable Crop Production
Article Type: Research paper
Corresponding Author: Drs. Linda Alrayes & Raymond Thomas
All Authors: Abraham Armah, Linda Alrayes, Thu Huong Pham , Muhammad Nadeem , Owen Bartlett, Eric Fordjour, Mumtaz Cheema, Lakshman Galagedara, Lord Abbey, Raymond Thomas
Dear Reviewer,
I am grateful for the opportunity to submit a revised version of my manuscript, titled " Integrating Rock Dust and Organic Amendments to Enhance Soil Quality and Microbial Activity for Sustainable Crop Production." I would like to express my sincere thanks for the time and effort invested in providing valuable feedback on my work. I appreciate the insightful comments and suggestions provided by the reviewers, and I have carefully revised the manuscript to address all of their concerns. Please find below a point-by-point response to the comments and suggestions.
Comment 1:, “Formulated scientific hypothesis checked during the research is fine. Please, add it in the text. In addition the aim of the research should be corrected and connected with the experiment conducted.”
Response 1: Thank you for the feedback. I have addressed your comment by incorporating the scientific hypothesis directly into the text and ensuring that the aim of the research is clearly stated and aligned with the experiment conducted.
Comment 2: “In abstract, add methods used, total number of samples, repetition and p-value”.
Response 2: Thank you for the comment. I have incorporated the requested details into the abstract, including the methods used, total number of samples, number of repetitions (n=4), and the significance level (p=0.05).
Comment 3:” In section 2.1. add description of Promix and Huplaso. What is it? I do not know such substances. Add key agronomic, physical or chemical properties. At least, in Supplementary. Please, briefly describe the key properties of other substances. Topsoil – soil name, sampling site, size of sample etc. Compost – what kind, sampling site, manufacturer, key properties. Biochar – what kind, manufacturer, key properties. (I did not find this valuable information in the section "Nutrient Composition of RD and the Formulated Amendments".)
Response 3: Thank you for your comment. The descriptions and key properties of Promix, Huplaso, topsoil, compost, and biochar, including their sources, compositions, and relevant agronomic or chemical characteristics, have been added under Section 2.2.
Comment 4: “Regarding the comment, "Please provide a map of soil sampling points,"….’ Please, add this information in the text”.
Response 4: Thank you for your comment. The information regarding the soil sampling points has been added to the text under Section 2.2 along with the soil description.
Comment 5: “For the comment, "Methods’…’ Please, justify using ANOVA and Pearson’s correlation analysis by the results of a test for normality (in section 2.9, write ‘all subsamples are fit normal distribution’ or ‘ANOVA and Pearson’s correlation analysis was applied after log-transformation to fit normal distribution’ [or something like this])”.
Response 5: Thank you for your comment. A statement has been added under Section 2.9 indicating that all subsamples conformed to a normal distribution prior to analysis. This justifies the use of ANOVA and Pearson’s correlation analysis.
Comment 6: “Please let me know if you would like me to submit the normality test results once access is available’. Yes, it needs”.
Response 6: Sounds good.
Comment 7: “Comment 4 and Response 4. Nice! Please, add this limitation in Methods or in Discussion with the regulatory standards used in supplementary table”.
Response 7: Thank you for your comment. The noted limitation has been added to the Discussion section.
Comment 8: “L.128 ‘The basic physical properties of RD’. Highlight it in supplementary table”.
Response 8: Thank you for your comment. The basic physical properties of RD are presented in Table 2.
Reviewer 2 Report
Comments and Suggestions for Authors
Dear Authors,
Thank you for considering my recommendations and working on implementing them in the manuscript body. Upon re-reading the manuscript, I noticed some minor areas that could still be improved. Please find my recommendation below:
L33: release “(28.36±4.88%) compared to control P (9.49±0.06%).” with “(28 %) compared to control P (9 %).” The decimals/sd - they load the reader and have no sense in this section
L328: Please explain what it means when the microbiome profile is expressed in %. Given the statement in L180-182, why did the author choose to present the results in % instead of quantitative mode?
L319: Figure 2.A. The authors should improve the mark sizes of the graph – and also make some colour adjustments for eq., replace yellow colour with other more easily observable
L405-414: Should be well developed and discussed in the discussion section
Considering the points as mentioned above, I recommend that the current version of the manuscript undergo further improvements before it is considered for publication in the Plants journal.
Best regards
Author Response
Answers to the Reviewer
Manuscript ID.: plants-3518145
Title: Integrating Rock Dust and Organic Amendments to Enhance Soil Quality and Microbial Activity for Sustainable Crop Production
Article Type: Research paper
Corresponding Author: Drs. Linda Alrayes & Raymond Thomas
All Authors: Abraham Armah, Linda Alrayes, Thu Huong Pham , Muhammad Nadeem , Owen Bartlett, Eric Fordjour, Mumtaz Cheema, Lakshman Galagedara, Lord Abbey, Raymond Thomas
Dear Reviewer,
I am grateful for the opportunity to submit a revised version of my manuscript, titled " Integrating Rock Dust and Organic Amendments to Enhance Soil Quality and Microbial Activity for Sustainable Crop Production." I would like to express my sincere thanks for the time and effort invested in providing valuable feedback on my work. I appreciate the insightful comments and suggestions provided by the reviewers, and I have carefully revised the manuscript to address all of their concerns. Please find below a point-by-point response to the comments and suggestions.
Comment 1: L33: release “(28.36±4.88%) compared to control P (9.49±0.06%).” with “(28 %) compared to control P (9 %).” The decimals/sd - they load the reader and have no sense in this section
Response 1: Thank you for your comment. That is absolutely correct, I have revised the values as suggested.
Comment 2: “L328: Please explain what it means when the microbiome profile is expressed in %. Given the statement in L180-182, why did the author choose to present the results in % instead of quantitative mode?”
Response 2: Thank you for your comment. The microbiome profile is expressed in percentages to represent the relative abundance of microbial taxa within each sample. This method allows for standardized comparison across treatments, regardless of variation in total read counts. Presenting the data in percentage format is a common practice in microbiome studies, as it effectively highlights shifts in community composition.
Comment 3: “L319: Figure 2.A. The authors should improve the mark sizes of the graph – and also make some colour adjustments for eq., replace yellow colour with other more easily observable”
Response 3: Thank you for your comment. I have replaced the yellow color in Figure 2.A with a more easily distinguishable alternative to improve visibility.
Comment 4: L405-414: Should be well developed and discussed in the discussion section
Response 4: Thank you for your comment. We have further developed and discussed the content from lines 405–414 in the Discussion section as suggested.
Round 3
Reviewer 1 Report
Comments and Suggestions for Authors
All corrections needed were made.